# Human-like driving behaviour emerges from a risk-based driver model

Sarvesh Kolekar [1✉], Joost de Winter [1] & David Abbink [1]

Current driving behaviour models are designed for specific scenarios, such as curve driving, obstacle avoidance, car-following, or overtaking. However, humans can drive in diverse scenarios. Can we find an underlying principle from which driving behaviour in different scenarios emerges? We propose the Driver's Risk Field (DRF), a two-dimensional field that represents the driver's belief about the probability of an event occurring. The DRF, when multiplied with the consequence of the event, provides an estimate of the driver's perceived risk. Through human-in-the-loop and computer simulations, we show that human-like driving behaviour emerges when the DRF is coupled to a controller that maintains the perceived risk below a threshold-level. The DRF model predictions concur with driving behaviour reported in literature for seven different scenarios (curve radii, lane widths, obstacle avoidance, roadside furniture, car-following, overtaking, oncoming traffic). We conclude that our generalizable DRF model is scientifically satisfying and has applications in automated vehicles.

[1] Department of Cognitive Robotics, Faculty of Mechanical, Maritime and Materials Engineering (3mE), Delft University of Technology, Mekelweg 2, 2628 CD Delft, The Netherlands. ✉email: s.b.kolekar@tudelft.nl

With the introduction of automated vehicles, humans will increasingly need to interact with automated systems. One of the factors that influence human-automation interaction is the trust that users have in the system[1,2]. Research suggests that the more technology seems to have human-like capacities, the more people are expected to trust it to perform its intended function competently[3]. For example, when recorded vehicle trajectories were played back to drivers, the drivers preferred a driving style they thought was their own[4]. To impart human-like capabilities in automated systems, understanding and modelling the human driver is essential.

Despite many efforts in the field of driver modelling (for surveys, see refs. [5–7]), driver models are typically developed for specific scenarios. For example, longitudinal behaviour has been modelled using the optical edge rate on open roads[8], the time to extended tangent point in curves[9], time to collision (TTC) while approaching obstacles[10] and time headway (THW) during car following[11]. Lateral positioning has been modelled using two-point (i.e., anticipatory vs. compensatory) models[12,13] in normal driving, and open-loop steering corrections[14] in emergency scenarios. To the best of our knowledge, the literature does not include a model of human driver behaviour that is applicable to a multitude of scenarios.

Practically, a unitary model could be developed by including a switch that selects a sub-model (or model parameters) based on the current driving scenario. However, this would require a priori identification of all possible scenarios, linked to appropriately parameterized models, and smooth transitions between them. Such an approach has two main problems: Firstly, the fragmented approach will not perform satisfactorily for driving situations where there is an inappropriate switch between tasks, or for driving situations that have not been addressed a priori, a problem also reported for machine learning techniques[15]. Secondly, this fragmented approach is not scientifically satisfying since it does not elucidate the underlying principles governing driving behaviour. These principles can be seen as a 'cost function' that human drivers try to minimise. Such cost functions have been proposed in the area of human motor control and have demonstrated emergent motor-control behaviour in different tasks and environments[16]. The present paper explores whether a similar generalizable model can be made for driving in different scenarios.

Essential to generalizable models is a cost function that is based on existing theories that aim to explain driving behaviour in a unified manner. The first attempt to such a unified theory was made by Gibson and Crooks[17]. They proposed that drivers perceive the qualitative concept of a 'field of safe travel', which is comprised of the possible paths that the car can take unimpeded. This theory paved the way for 'motivational driver models' such as the risk homoeostasis and task-difficulty homeostasis theories by Wilde[18] and Fuller[19], respectively. However, these theories have two drawbacks: Firstly, they lack specificity regarding their internal mechanisms, which makes it difficult to operationalize and validate them[20–22]. Secondly, homeostasis theories cannot account for an important characteristic of human-driving behaviour, namely satisficing. Drivers do not optimise their states (e.g., they do not try to follow the centreline of the road perfectly) but try to maintain their state within acceptable limits (e.g., within lane boundaries)[23]. Models based on homeostasis theories maintain a certain risk or task-difficulty level, and hence will always follow a reference trajectory (for example, centreline of the road), which is not coherent with satisficing behaviour.

Näätänen and Summala[24] addressed satisficing behaviour by introducing the concept of a risk-threshold. According to their theory, drivers do not maintain a certain level of risk but make corrective actions only when the risk they perceive increases beyond a threshold. This means that any vehicle state is acceptable, as long as the driver's risk is within his/her individualised threshold. However, to the best of our knowledge, the risk-threshold theory has not been operationalized and tested in different driving scenarios.

In this paper, we propose a novel risk metric, based on published empirical data, that operationalizes the risk-threshold theory. We then formulate a driver model that utilises the proposed risk metric as a cost function, simulate it in different driving scenarios, and compare its predictions of driver behaviour with driver behaviour reported in literature. The results exemplify that, in driving, similar to motor-control tasks, a cost function that accounts for the consequence of noise (in human's perception and actions) seems to be the underlying principle governing driving behaviour. In short, we propose a risk metric that operationalizes human-like behaviour in a unified manner, for different driving scenarios.

## Results

**Quantifying perceived risk.** According to Näätänen and Summala[24], perceived risk is the product of the subjective probability that an event will occur and the consequence of that event (Fig. 1a). In this paper, we operationalize these components (Fig. 1b).

The consequence of an event is the dangerousness of being in a particular state. We quantified this by assigning a cost to objects in the driving scene according to the danger they pose. These values need to be identified experimentally and are independent of the driver. A representation of the driver's subjective belief about the probability of an event occurring was quantified by Kolekar et al.[25]. They measured drivers' subjective (self-reported) risk levels and objective (steering angle) steering responses in an obstacle avoidance task. The Driver's Risk Field (DRF), as Kolekar et al.[25] called it, has a high value near the ego car and decays as the lateral and longitudinal distance from the ego car increases. The DRF hence indicates that the driver believes that there is a higher probability of being in a position near their current position, in the next $t_{la}$ seconds (preview time), than at further away points. The DRF, in essence, captures the driver's uncertainty in his/her perception and actions.

The quantified perceived risk (risk metric) is a scalar value which is the product of the 'cost of an event' and the DRF, summed over all the grid points. In essence, this risk metric quantifies the 'consequence of noise/variability in our perception and actions', which is similar to the unifying cost functions proposed in motor control[16,26].

**Modelling the DRF.** The DRF has been previously quantified for a fixed speed on a straight road[25]. In this section, we provide the mathematical formulation of a DRF that moves with the driver and changes its shape with the speed and steering angle. In this paper, the predicted vehicle path is calculated using a kinematic car model. The position ($x_{car}$, $y_{car}$), heading ($\phi_{car}$), and steering angle ($\delta$) determine the radius of the arc ($R_{car}$) in which the car is predicted to travel, assuming a constant steering angle (Eq. (1)).

$$R_{car} = \frac{L}{\tan(\delta)} \qquad (1)$$

$L$ is the wheel-base of the car. Using $x_{car}$, $y_{car}$, $\phi_{car}$ and $R_{car}$, the centre of the turning circle ($x_c$, $y_c$) is determined, which is used to calculate the arc length ($s$), measured along the predicted path (Fig. 2a).

The DRF is modelled as a torus with a Gaussian cross-section (Eq. (2)). The height ($a$) and width ($\sigma$) of the Gaussian are a

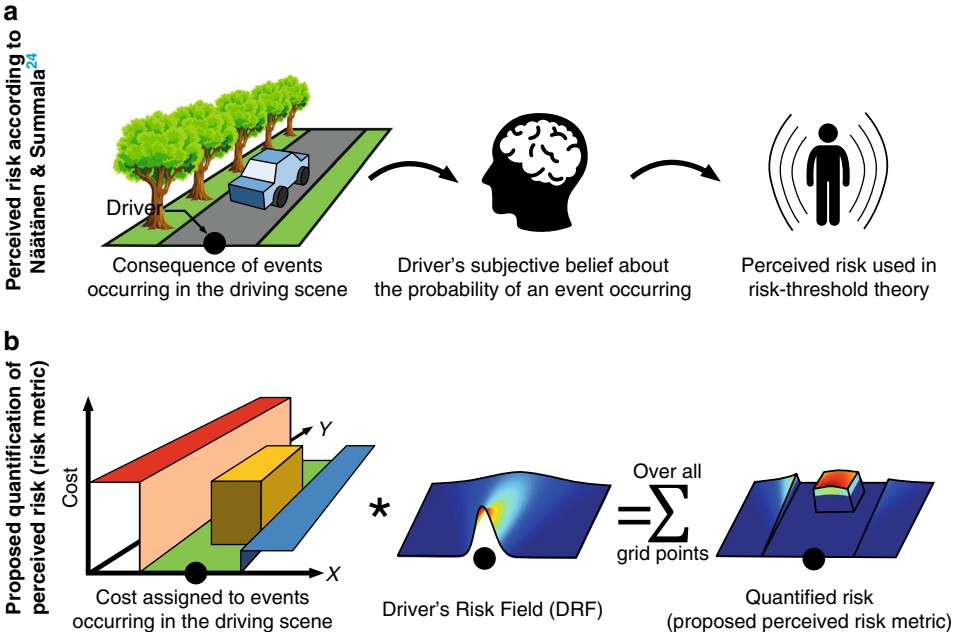

**Fig. 1 Visualising the quantification of driver's perceived risk. a** This row illustrates Näätänen and Summala's[24] formulation of perceived risk. The consequence of an event (e.g., colliding with a tree) and the driver's subjective belief about the probability of that event occurring, form the driver's perceived risk. The driver in the ego car is indicated using the black marker. **b** This row illustrates the proposed quantification of this perceived risk. The cost of each element in the driving scene is multiplied with the Driver's Risk Field (DRF) that represents the driver's belief of the probability of being in a position. This product summed over all grid points generates the estimate of quantified risk.

function of the arc length ($s$) (Fig. 2b).

$$z(x,y) = a \exp\left(\frac{-\left(\sqrt{(x-x_c)^2 + (y-y_c)^2} - R_{car}\right)^2}{2\sigma^2}\right) \quad (2)$$

The height of the Gaussian ($a$), is modelled as a parabola (Eq. (3)).

$$a(s) = p(s - v t_{la})^2 \quad (3)$$

With a fixed look-ahead time ($t_{la}$), the look-ahead distance is assumed to increase linearly with speed ($v$). Parameter ($p$) defines the 'steepness' of the parabola.

The width of the Gaussian ($\sigma$) is modelled as a linear function of arc length ($s$) (Eq. (4)), which is a simplification of the parabolic function (Supplementary Fig. 1) used in Kolekar et al.[25] and includes the following parameters: first, $c$ defines the width of DRF at the location of the vehicle and is related to the car-width. In this paper, $c$ is equal to car-width/4 ($\pm 2\sigma$ of Gaussian distribution accounts for 95%). Second, $m$ defines the slope of widening (or narrowing for negative values) of the DRF when $\delta = 0$ (driving straight). Third, $k_1$ and $k_2$ increase (or decrease, for negative values) the width of the DRF proportional to the (absolute) steering angle ($|\delta|$). This is based on the rationale that variability in steering angle increases linearly with the steering angle[11,27]. It is similar to the empirically confirmed signal-dependent noise present in the human sensorimotor system[26,28]. $k_1$ and $k_2$ represent the parameters for the inner and outer edges of the DRF, respectively, and allow for an asymmetric DRF. The expansion of DRF proportional to $\delta$ results in the accumulation of a higher risk for a curve with a smaller radius. The asymmetric expansion ($k_1$ and $k_2$) provides flexibility to exhibit curve-cutting ($k_1 < k_2$), centreline ($k_1 = k_2$), or curve overshooting ($k_1 > k_2$)

behaviour.

$$\sigma_i = (m + k_i|\delta|)s + c \quad (4)$$

$$i = 1(\text{inner } \sigma), 2(\text{outer } \sigma)$$

In short, the DRF is parameterized by $p$, $t_{la}$, $m$, $c$, $k_1$, $k_2$, and is only dependent on driver's state, not the environment.

To test if the proposed risk metric can operationalize human-like behaviour in a unified manner, we used the risk metric as an input for a simple driver model ('Methods' section) and simulated it on a virtual track (Fig. 3a). The main characteristic of the DRF driver model is that it does not minimise the cost function. Instead, it tries to achieve a certain goal (in this paper, a desired speed $V_{des}$), while maintaining the cost (quantified perceived risk: $C$) below an individualised threshold ($C_t$).

To identify realistic parameter values for the driver model, we replicated the track used to simulate the model (Fig. 3a), in a driving simulator. A 25-year-old male volunteer drove ten times with the instruction, 'drive as you normally would' and ten times with 'drive faster'. This was meant to emulate 'normal' and 'sport' driving behaviour. A section of the track (Fig. 3) was used for parameter estimation. The speed and lateral deviation trajectories estimated by the DRF model showed a close resemblance to those of the participant who also drove faster in sport setting than in normal setting. Also, the trajectories remained, for most parts, within the $\pm 2\sigma$ bound of the human trajectories. These results show that the DRF driver model can operationalize driving behaviour and remain within the human-like trajectory bounds ($\pm 2\sigma$). These were necessary, but not sufficient checks. To verify if the proposed quantified risk is indeed human-like, we compare the predictions of the DRF model with the results published in human-driving behaviour studies in literature.

**Validation using published literature**. To validate the DRF model, we selected papers from literature that investigated driver

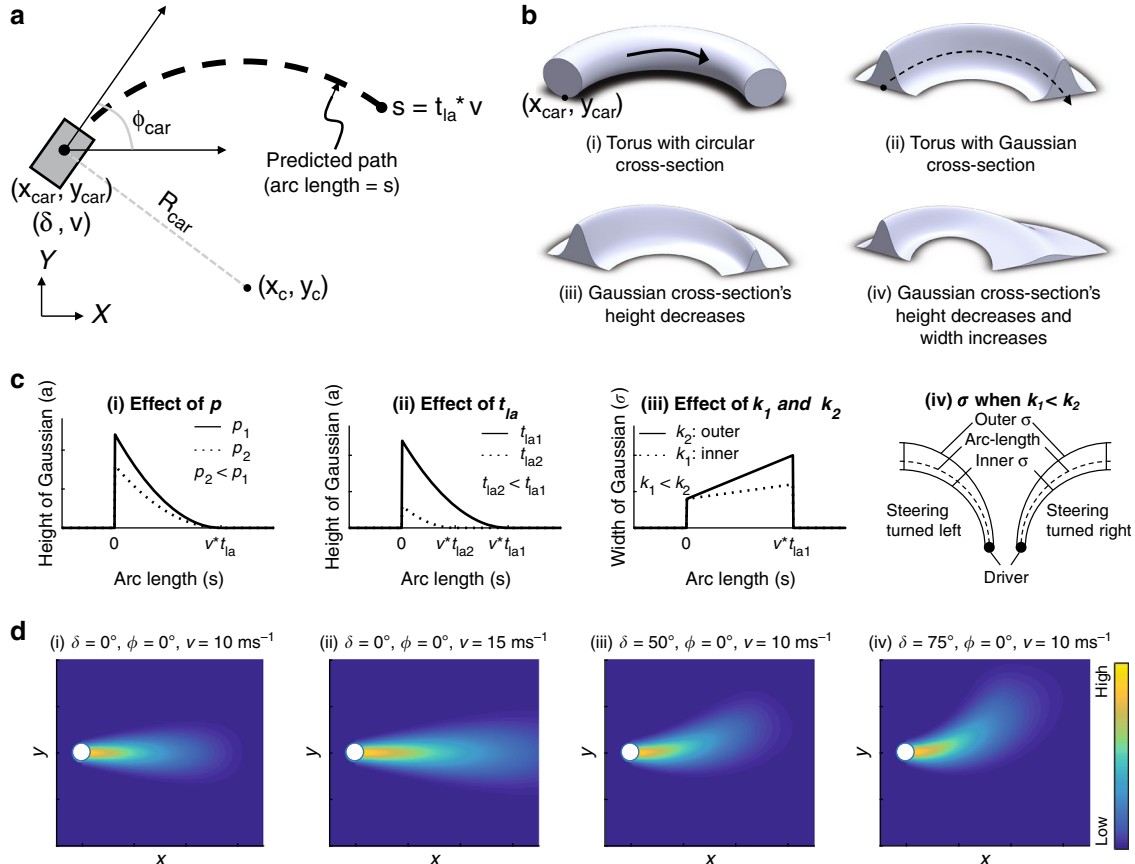

**Fig. 2 Modelling the Driver's Risk Field. a** The 'predicted path' is calculated using the trajectory of vehicle kinematics, assuming constant steering angle ($\delta$) and speed ($v$) for a fixed look-ahead time ($t_{la}$). **b** The DRF is modelled as a modified torus. Four steps are taken to form the DRF from (i) A torus that curves along the 'predicted path'. (ii) Cross-section of torus is modified to a Gaussian. (iii) Height ($a$) and (iv) width ($\sigma$) of the Gaussian become functions of arc length ($s$), Eqs. (3) and (4), respectively. **c** The DRF is parameterized by six parameters: $p$, $t_{la}$, $k_1$, $k_2$, $m$, $c$. The effect of $p$ (steepness of the parabola) and $t_{la}$ are shown in (i) and (ii) and emerge from Eq. (3). The maximum height of the Gaussian is determined by $p$, $t_{la}$ and speed. (iv) Parameters $k_1$ and $k_2$ link the steering angle to the width of the Gaussian. The DRF widens (if $k_1$, $k_2 > 0$) or narrows (if $k_1$, $k_2 < 0$). (v) $k_1$ and $k_2$ correspond to the inner and outer Gaussian widths, respectively. So, if $k_1 < k_2$, the inner Gaussian is narrower than the outer, which enables 'corner cutting' in curves. **d** The figure shows the shape and magnitude of DRF as a function of the position in the driving scene (global $x$ and $y$ coordinates). The DRF is a dynamic field that expands with an increase in speed (compare (i)–(ii)) and steering angle (compare (i)–(iii) and (iii)–(iv)). MATLAB code for DRF GUI is provided in the Code availability section.

behaviour as a function of road and traffic conditions in terms of speed and lateral position. Since no single study fully replicates our scenarios, we chose different studies from literature, to compare with the respective DRF model predictions. Wherever possible, we chose a naturalistic driving study in similar conditions as simulated.

**Effect of road scenarios**. We tested four road scenarios: different curve radii, different lane widths, obstacle avoidance and roadside furniture.

**Curve radius**. The effect of curve radius on driving behaviour was examined by investigating the lateral position (curve-cutting behaviour) and speed while driving through curves.

Lateral position: Research has shown that drivers exhibit 'curve-cutting', that is, they do not follow the centreline of the lane but try to increase the effective radius of travel[29–31]. For model validation, we selected the on-road study by Xu et al.[32] because it provides the largest sampling of curve radii (0–200 m). They found that the amount of curve-cutting reduced as the curve radius increased (Fig. 4-1b), which is coherent with the predictions of the DRF driver model (Fig. 4-1a). They quantified

curve-cutting behaviour using the trajectory transection rate (TTR), which normalises the lateral deviation from the lane centre with respect to the lane width, in curves. The DRF model exhibits curve-cutting behaviour due to its asymmetric shape defined by parameters $k_1$ and $k_2$ (Fig. 2c). The DRF model also predicts that curve-cutting is higher in sport setting than in normal setting.

Speed: Several studies report that the speed at which a curve is taken increases non-linearly with curve radius, in driving simulator[33,34] and on-road tests[11,33,35]. The paper from Taragin and Leisch[36] was chosen (Fig. 4-1d) because their on-road study provided data on curve radii range (60–714 m) and lane width range (2.6–4.3 m), which are similar to that simulated for the DRF model. The DRF model predicts that the speed increases with curve radius, asymptotically approaching straight road speed for a large radius (Fig. 4-1c), which is similar to the experimental results of Taragin and Leisch[36] (Fig. 4-1d). The DRF model exhibits this speed dependency on curvature because the width of the DRF changes with steering angle (Eq. (4)).

**Lane width**. The effect of lane width was examined using the standard deviation of lateral position (SDLP) and speed.

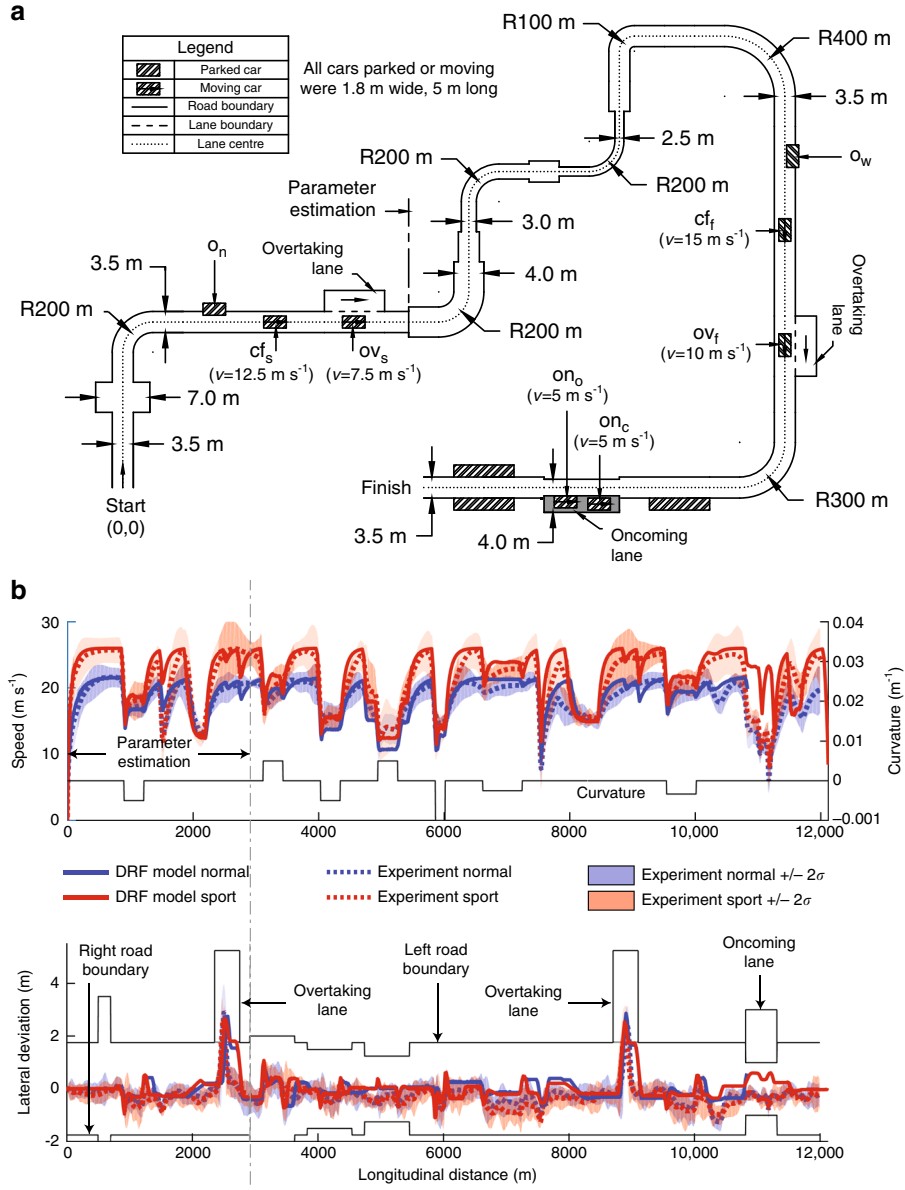

**Fig. 3 Track used for testing the driver model. a** The track contains four road and three traffic scenarios. The four road scenarios are (1) Curve radii: R100m, R200m, R300m and R400m, (2) Lane widths: 2.5, 3.0, 3.5 and 4.0 m, (3) Obstacle avoidance: A car was parked on a 3.5-m wide straight section such that 0.9 or 1.4 m of the car-width encroached on the road to simulate narrow ($o_n$) and wide ($o_w$) obstacles, respectively. (4) Roadside furniture: A 200-m long row constituting of 10 cars was placed either outside the left lane boundary (asymmetric) or outside both lane boundaries (symmetric). The three traffic scenarios are (1) Car following: Two cars travelling at a constant speeds of 12.5 m s$^{-1}$ ($cf_s$) and 15 m s$^{-1}$ ($cf_f$) along the lane centre on different straight sections were followed. (2) Overtaking: Two cars travelling at constant speed of 7.5 m s$^{-1}$ ($ov_s$) and 10 m s$^{-1}$ ($ov_f$) were overtaken using a 3.5 m overtake lane. (3) Oncoming traffic: Two cars travelling at a constant speed of 5 m s$^{-1}$ on the 2-m wide oncoming lane, approached the ego car. The first oncoming car drove on the lane centre ($on_c$). The second car was offset 0.3 towards the ego car. **b** To identify realistic values for the parameters of the DRF driver model, we replicated the track in a driving simulator and one volunteer drove 10 times 'normally' (blue) and 10 times in a 'sporty manner' (red). Speed and lateral deviation from the lane centre are plotted as a function of the distance travelled along the lane centre of the track. The speed and lateral deviation trajectories of the DRF driver model, for the most part, lie within the $\pm\sigma$ limits of the experimental trajectories. The 'sport' parameter setting consistently drives faster than the 'normal' setting and in both cases shows similar trends in acceleration braking as shown by the human. The driver model maintains itself within the lane boundaries, while exhibiting satisficing (i.e., not always following the lane centre), even in conditions that were not experienced during parameter estimation.

Lateral position: SDLP, which represents the swerving behaviour of a car, is reported to increase with lane width, in a simulator study by Godley et al.[37]. They examined the SDLPs of participants on three different lane widths (2.5, 3.0, 3.6 m) (Fig. 4-2b). Similar results are reported in other simulator[38,39] and on-road studies[40] which are coherent with the predictions of the DRF model (Fig. 4-2a). On a wider road, the DRF model has wider areas of low cost and hence, can use a larger width of the road without steering corrections (exhibit satisficing), resulting in higher SDLP.

Speed: It is reported that the speed at which drivers negotiate roads increases as the lane width increases, in simulator[37,41–43] and on-road studies[40,44]. The DRF model also showed a similar increase in speed with lane width (Fig. 4-2c) and is compared to

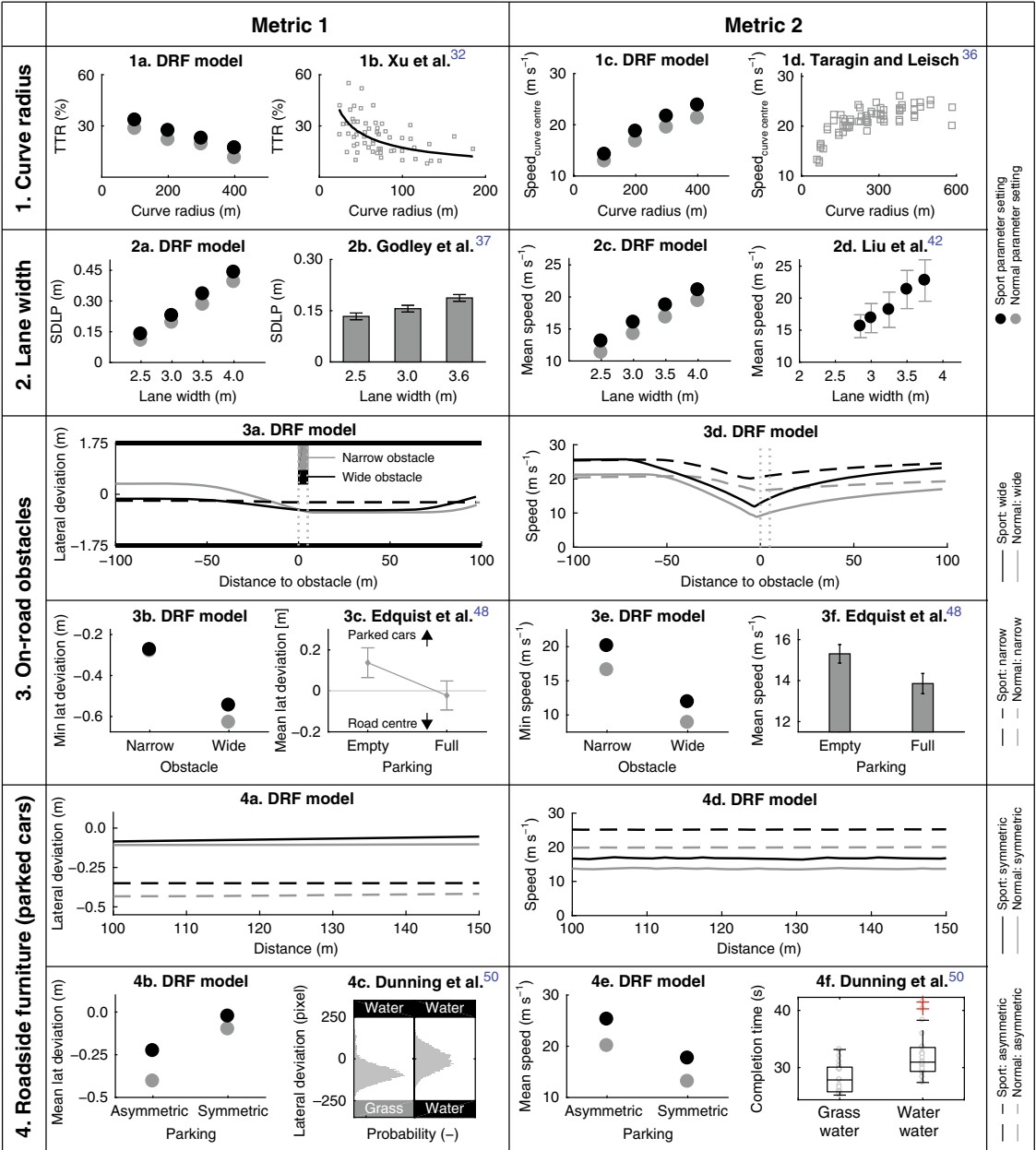

**Fig. 4 Validating the model in road scenarios using literature.** Each row represents one scenario and the columns compare two different metrics in that scenario. The DRF model results are compared to the results from literature (Supplementary Notes 1–8) in the adjacent subfigures (Supplementary Figures 3–6). In the DRF model subfigures, the black and grey markers represent the sport and normal parameter settings, respectively. **1** Curve radius: 1a and 1b show that the DRF model predicts the decrease in 'curve-cutting' (quantified using TTR) as curve radius increases. 1c and 1d show the speed at the curve centres. The sport setting of DRF cuts the curves more (1a) and drives at higher speeds (1c) compared to the normal setting. **2** Lane width: 2b shows that the (mean ± SE) standard deviation of lateral position (SDLP) of the vehicle increases as the lane width increases. The DRF model (2a) can predict this trend. 2c and 2d (mean ± SD) show that the speed at which drivers negotiate a road increases as the lane width increases. **3** On-road obstacles: In 3b, the 'wide' obstacle encroaches more onto the road compared to the 'narrow' obstacle. The minimum lateral deviation (3b) is calculated from the trajectories in 3a. Drivers moved away from the parked cars (3c: lane centre = 0, bars indicate 95% CIs). 3b shows that the DRF model showed a similar trend of moving away from the obstacle. Drivers drove slower when there were parked cars, as compared to when there were no parked cars encroaching the road (3f: bars indicate 95% CIs). 3e shows that the DRF model slows down for obstacles covering the road partially. **4** Roadside furniture: In the asymmetric case, the mean lateral deviation from the lane centre is away from the parked cars (4b) and away from water (more dangerous than grass) in 4c. Subfigure 4c shows the distribution of lateral position of the participants. 4e and 4f show that in the symmetric condition with 'danger' on both sides of the lane, the DRF model correctly predicted that the drivers drove slower than in the asymmetric case. The mean lateral deviation (4b) and mean speed (4e) are calculated from the trajectories in 4a and 4d, respectively.

the results from a (moving base) simulator study of Liu et al.[42]. On a wider road, there is a larger area of 'no risk', which means that the model can reach higher speeds before exceeding the risk threshold.

**On-road obstacles**. Obstacle avoidance was simulated for the DRF model by parking cars partially on the road, which led to a temporary 'narrowing' of the street. The effect of this temporary narrowing was examined by analysing the lateral deviation and

speed of the ego vehicle. Several researchers have reported, in on-road studies, that on-street parking induces 'traffic calming' by reducing the average speed[45–47]. We selected the simulator study of Edquist et al.[48] because they measured the effect of on-street parking on lateral position and speed.

Lateral deviation: Edquist et al.[48] reported that the mean lateral position of the vehicles shifted away from the parked cars (Fig. 4-3c). The DRF model yields a similar trend, where the ego car deviates away from the parked car (Fig. 4-3b).

Speed: A reduction in mean speed was reported in the presence of parked cars (Fig. 4-3f)[48], which is coherent with the behaviour shown by the DRF model (Fig. 4-3e). It should be noted that Edquist et al.[48] reported the mean speed since they had a row of parked cars. However, we had only one parked car, which means we can only report the minimum speed. The DRF model successfully avoided on-road obstacles by steering and braking.

**Roadside furniture.** Road shoulders, guard-rails, vegetation and parked cars have been reported to affect a vehicle's lateral position and speed[31,49]. The DRF model was simulated in an 'asymmetric' case where a 200-m long row of cars was parked outside the left lane boundary, and a 'symmetric' case where they were parked outside both lane boundaries. Dunning et al.[50] examined 'asymmetric' (with water (more risk) on one and grass (less risk) on the other side of the lane boundary), and 'symmetric' (with water on both sides) conditions in their experiment.

Lateral position: Dunning et al.[50] reported that the lateral position of the participants shifted towards the less dangerous grass in the asymmetric case and remained in the centre in the symmetric case (Fig. 4-4c). Similar results are seen in the behaviour of the DRF model, where the ego car moves away from the parked cars (at lateral position = +2.75 m) and remains in the centre of the lane in the symmetric case (Fig. 4-4b).

Speed: Dunning et al.[50] reported that participants, on average, drove slower in the symmetric case (Fig. 4-4f). The DRF model also shows similar behaviour where the ego car drove faster in the asymmetric case as compared to the symmetric case. This is because in the asymmetric case, the DRF model steered away from the 'risky' parked cars and could maintain a higher speed without exceeding the risk threshold. In the symmetric case, driving on the centreline was not enough to reduce the risk below the threshold and hence the model had to slow down. In both conditions, the sport setting drove faster than the normal setting of the DRF model. The DRF model could react to roadside furniture by steering and braking since the DRF spreads beyond the lane boundaries.

**Effect of traffic scenarios.** We tested three traffic scenarios, namely: car following, overtaking and interaction with oncoming cars.

**Car following.** We tested the effect of lead car speed on time headway (THW) and braking intensity during car following. We simulated 'slow' and 'fast' car following with lead cars that maintained constant speeds of 12.5 and $15 \, \mathrm{m \, s^{-1}}$, respectively.

THW: THW during car following represents the time available to the driver of the following vehicle to reach the same level of deceleration as the lead vehicle, in case the lead vehicle brakes. Several studies in literature examined the effect of lead vehicle speed on THW[51–53] and reported that (for lead car speed above $10 \, \mathrm{m \, s^{-1}}$) the preferred time headway under steady-state car following ($\mathrm{THW_{pref}}$) is almost constant and independent of the lead car speed. The DRF model also predicts an almost constant $\mathrm{THW_{pref}}$ (Fig. 5-1b). The DRF model, with the current parameter values, behaved more conservatively (higher $\mathrm{THW_{pref}}$) than the

average human driver, as reported by He et al.[53] in their on-road study (Fig. 5-1c). In addition, the $\mathrm{THW_{pref}}$ for the sport parameterization was smaller than that for the normal para-meterization of the DRF model. This concurs with the findings in the literature, where sensation-seeking drivers were reported to maintain lower $\mathrm{THW_{pref}}$ compared to sensation avoiding individuals[52,54].

Braking intensity: Another aspect of car following that is widely studied is the braking intensity of the car in response to the separation to the lead car. In a test-track study, Van der Horst[55] reported that the braking intensity (deceleration at the onset of braking) increased as the approach speed increased (Fig. 5-1f), which corresponds to the DRF model's results (Fig. 5-1e). The study also reported that with 'hard braking' instruction, participants' braking intensity was higher than in normal braking condition. The DRF model also predicts that a sport parameter setting (black markers) will yield higher deceleration than the normal setting (grey markers: Fig. 5-1e). The DRF model exhibits this behaviour since the lead car encroached the DRF at a higher rate when the approach speed was high and at a lower rate when the approach speed was low. This 'rate of encroachment' translated into velocity reduction at a proportional rate.

**Overtaking.** We studied the effect of lead vehicle speed on overtake-distance (distance covered during the overtaking man-oeuvre) and on the TTC at which the overtaking manoeuvre is initiated. To test the DRF model, we simulated a 'flying overtake manoeuvre' in which there are no oncoming cars on the adjacent lane. Figure 5-2a illustrates one of the major drawbacks of the DRF model: it overtakes the car but does not return to its own lane after the overtake. This is the drawback of using a cost-threshold-based satisficing controller. Since the model perceives the road to be twice as wide (ego + overtaking lane), it comes back (to its lane) just enough to bring the risk below its threshold (satisficing). Secondly, the DRF model would not be able to perform an 'accelerative overtake' since its speed is limited by the $V_{\mathrm{des}}$ parameter.

Overtake-distance: Crawford[56] reported that the overtake-distance increased with the speed of the overtaken car (Fig. 5-2c). This corresponds to the DRF model's behaviour, where the overtake-distance was higher for the $10 \, \mathrm{m \, s^{-1}}$ overtaken car than for the $7.5 \, \mathrm{m \, s^{-1}}$ overtaken car. In addition, note that the sport setting of the DRF model had larger overtake-distances than the normal setting.

TTC at overtake initiation: Several studies investigate time-to-collision (TTC = ratio of relative distance to relative speed) at the initiation of overtaking manoeuvres either to the lead car[57] or with the oncoming car[58,59] (outside of the scope of our scenarios). The on-road study by Chen et al.[57] reported that the TTC at (start of) lane change increased with the speed of the overtaken car (Fig. 5-2f). Similar behaviour is shown by the DRF model, but more interestingly, the sport setting of the DRF model maintained a lower TTC than the normal setting. In a driving simulator study, Farah[60] reported that young male drivers, generally considered sporty drivers, had smaller TTCs at lane change than adults.

**Oncoming traffic.** We examined the effect of oncoming traffic's lateral position on the DRF model's choice for speed and lateral position. We simulated a narrow rural road with 2-m wide ego and oncoming lanes, without any barrier in between. Lewis-Evans and Charlton[41] reported that on a two-lane rural road, drivers drove more towards the road centre, in the absence of oncoming traffic. The DRF model exhibits similar behaviour, with a bias ($\approx 50$ cm) towards the road centre (Fig. 5-3b: 'absent' condition).

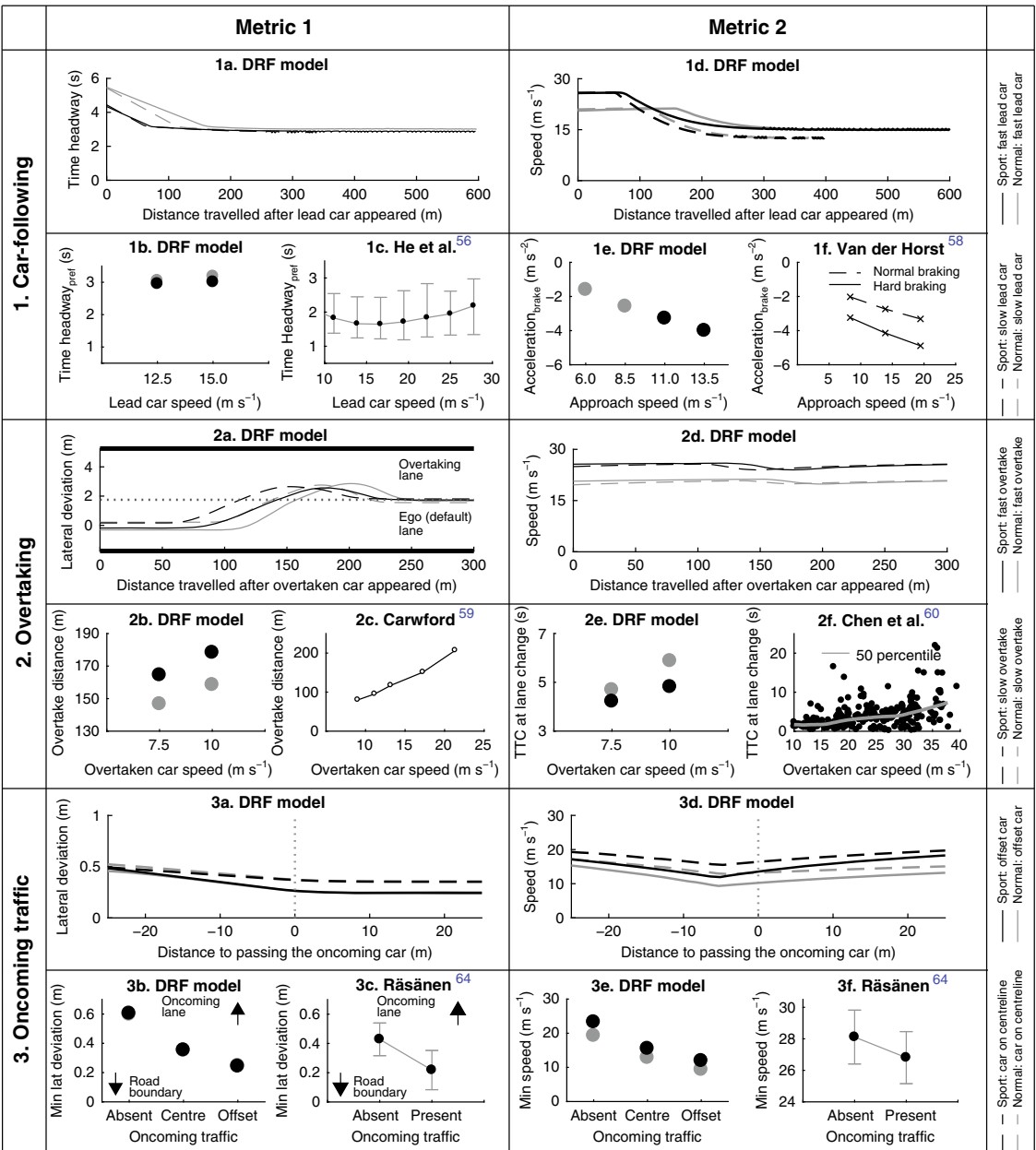

**Fig. 5 Validating the model in traffic scenarios using literature.** Similar to Fig. 4, each row represents one scenario and the two metrics in the two columns compare the DRF model results to trends shown in the literature (Supplementary Notes 9–14). For the DRF model figures, the black and the grey markers represent the sport and normal parameter settings, respectively (Supplementary Figs. 7–9). **1** Car following: 1b and 1c indicate that the preferred time headway is independent of the speed. In 1c, the circular markers indicate the median and the whiskers indicate 25th and 75th percentile. 1e and 1f show that the braking intensity (represented by the acceleration at brake initiation) increases as the approach speed to the obstacle increases. **2** Overtaking: 2b and 2c show that the DRF model could correctly predict that the overtake-distance increases as the speed of the overtaken car increases. In the sport setting, the model covers larger distance than in normal setting, indicating 'smoother' trajectories in the sport setting. However, the DRF model does not come back to its own lane sufficiently (2a). Subfigures 2e and 2f show that the predictions of the DRF model agree with the results in literature that show the time to collision (TTC) at the start of the overtake manoeuvre increases, as the speed of the overtaken car increases. **3** Oncoming traffic: In 3b and 3c, the minimum lateral deviation is shown on the y-axis. The condition where no oncoming cars were present is indicated by 'absent'. The DRF model simulated one car that drove on the oncoming lane's centre ('centre' in 3b) and another car that was offset towards the ego lane ('offset' in 3b). In normal and sport setting the DRF model moved away from the oncoming traffic, which is in agreement with the driver's behaviour. 3e and 3f show that the DRF model slowed down, like humans (3f), when it encountered oncoming traffic. In 3c and 3f, the black markers indicate mean, and whiskers indicate the ±SD.

The model shows this behaviour because the paved road to the left (i.e., oncoming lane with no traffic) is less 'dangerous' than the road boundary to the right.

Lateral position: Studies that investigated the effect of oncoming traffic[61–63] have reported that drivers' lateral position depends on the presence of oncoming vehicles in the adjacent lane. Rasanen[61], in an on-road study, compared driver's lateral position with and without oncoming traffic (Fig. 5-3c) and reported behaviour similar to DRF model predictions, where the lateral position moves away from the lane with oncoming traffic. In addition, it moves even further when the oncoming car is offset towards the lane position of the ego car (Fig. 5-3b).

Speed: The DRF model slowed down in the presence of oncoming traffic, and slowed down more when the lateral position of the oncoming car was offset towards the ego car (Fig. 5-3e). Rasanen[61] (Fig. 5-3f) reported no significant difference in speed between the oncoming traffic 'absent' and 'present' conditions. However, Rosey et al.[62] reported a significant reduction in speed when drivers encountered oncoming vehicles. Moreover, they also reported a significant decrease in speed while encountering trucks as compared to cars[62], which is in line with the predictions of the DRF model.

## Discussion

In this paper, we set out to find the underlying principle that governs human-driving behaviour, implement this into a cost function for an operational driver model, and evaluate the generalizability of the modelled behaviour across different traffic scenarios by comparing it to adaptations in speed and lateral position from available literature of real-world and driving simulator studies.

One of the principles that emerged from qualitative driver behaviour theories was 'perceived risk', However, to the best of our knowledge, 'perceived risk' has not been quantified or used in a driver model to generate human-like driving behaviour. In this paper, we operationalized the 'perceived risk' by multiplying the DRF (which accounts for the driver's perception-action uncertainty) with the cost map of the driving scene (which quantifies the consequence of a hazard/event). This makes the cost function 'uncertainty-aware'.

A driver's 'uncertainty-awareness' is embedded in the DRF model via four features. First, the DRF widens along the 'predicted path' and hence is wider than the car-width. Without this feature, the DRF model would not slow down on a narrow road (wider than car-width). Second, the DRF widens and elongates with increasing speed. Without this, the DRF model would not maintain constant time headways in car following or slow down

for curves. Third, the DRF widens with an increase in steering angle. Without this feature, the DRF model would not slow down more for curves with higher curvature than for curves with lower curvature, and would negotiate all the curves at the same speed. Fourth, the asymmetric widening of the DRF along the 'predicted path' (generally $k_1 < k_2$) lets the model exhibit 'curve-cutting' behaviour. Without the asymmetric widening, the model would always follow the lane centre.

Dealing with uncertainty in the ego-robot's and the external obstacles' location has been widely studied[64,65]. Several models, ranging from tentacle-like algorithms[66] to Rapidly-exploring Random Trees (RRT)[67], have been proposed for trajectory and speed planning. The methods that are closest to the cost function proposed in this paper are based on uncertainty propagation[68]. Most of these algorithms account for the first two points mentioned in the previous paragraph, namely: widening of the uncertainty with predicted path and speed dependency of uncertainty field. In addition, these algorithms account for the uncertainty in predicting the future location of the obstacles. This feature needs to be incorporated in the driving scene cost map of future versions of the DRF model (Fig. 6d). However, algorithms in the literature seldom incorporate the latter two features: widening of uncertainty with steering and asymmetric uncertainty propagation; hence, existing models cannot produce 'curve-cutting' and curvature-dependent speed negotiation, behaviours that are seldom required in robotic applications. In short, to generate human-like behaviour, the underlying cost function has to be 'uncertainty-aware' and incorporate the (motor-control inspired) effect of signal-dependent noise to replicate the speed-accuracy trade-off that we see in driving behaviour.

Implementing a satisficing controller in a potential field has its drawbacks. The model did not return to its lane after overtaking the lead car because it can sense hazard only from physical objects (e.g., cars, road boundary) and cannot perceive the 'tactical' risk of being in an oncoming lane. Other tactical risks, such as risks

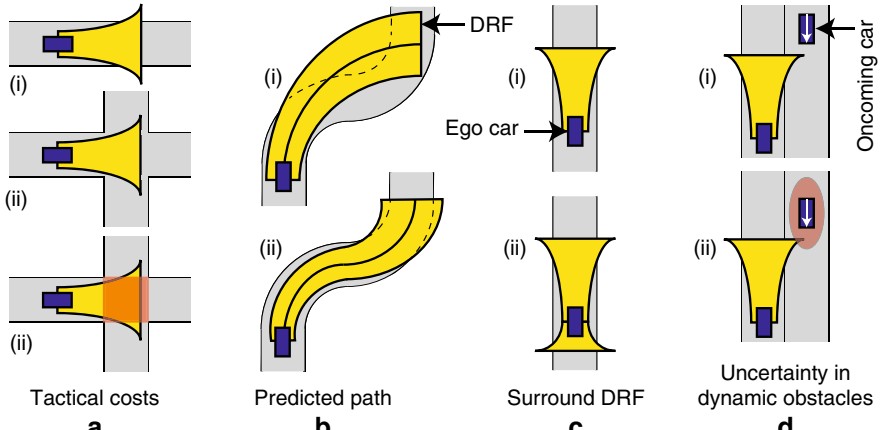

**Fig. 6 Limitations of the model. a** Tactical costs: The DRF model can only perceive physical risk from objects such as cars, trees, etc. However, it cannot perceive the risk from oncoming traffic which is currently not in its field of view. Hence, at an intersection, rather than slowing down, it will speed up, since there is larger road-area available, which is contrary to what a human would do. This can be solved by introducing additional 'tactical costs' that artificially increase the risk of an intersection (red square). This approach can be extended to other elements such as traffic lights or zebra crossings. **b** Predicted path: For simplicity, the DRF model currently uses a circular arc for predicting the path (for preview time $t_{la}$ seconds). This circular path arises due to the assumption that the current steering angle ($\delta$) and speed ($v$) will be held constant over the preview time. However, we can optimise for a vector of steering angles and speed (as is done in a Model Predictive Control). This allows for a flexible DRF and better prediction of microscopic trajectories. **c** Surround DRF: In this paper, the DRF only extends in front of the vehicle (top). However, the risk field extends on all four sides. The bottom image is merely a suggestion, and the shape has not been investigated. This 'surround DRF' will help predict human-driving behaviour in additional scenarios such as: being followed by another car, being overtaken, lane change manoeuvres, etc. **d** Uncertainty in dynamic obstacles: The DRF represents the driver's (self) perception-action uncertainty. However, the motion of dynamic obstacles is less predictable. This uncertainty was ignored in this paper, but will have to be accounted for in future iterations of this model.

that may occur when approaching an intersection or a red traffic light, are not incorporated in the model either. However, the structure of the model facilitates the addition of these 'tactical' costs to different road elements. Other limitations include the use of car-kinematic model, using a circular arc for 'predicted path' calculations, and the DRF extending only in front of the ego car. In future iterations, a car-dynamic model, a spline instead of a circular arc (Fig. 6b), and a DRF that surrounds the vehicle on all four sides (Fig. 6c) can help generate better microscopic trajectories and generate behaviour in more scenarios (e.g., ego car being overtaken).

Satisficing behaviour becomes important when developing advanced driver assistance systems (ADAS) that physically interact with the driver, e.g., the haptic shared controller (HSC)[69], which guides the driver via torques on the steering wheel. If the HSC tries to follow a reference (e.g., the lane centre), it will exert a torque and bring the driver to the centreline, even if the driver was satisfied with an off-centre lateral position. To avoid these undesired torques that can severely hamper the acceptance of the system, we need threshold-based models that can exhibit satisficing behaviour.

An important contribution of this paper is the extensive literature-based validation. Note that, in this paper, we do not compare the trajectories of steering angle, speed and lateral deviation, but assess the behaviour of the model by comparing trends in certain metrics to those reported in the literature. Six out of the seven scenarios were validated using on-road studies or studies from driving simulators backed by on-road studies (only simulator studies found for roadside furniture: Supplementary Tables 1–8). In Fig. 4 (road scenarios), owing to the simplicity and 'static' nature of road elements, there was abundant literature and consensus amongst researchers as to which metric reflected human behaviour (e.g., curve-cutting: TTR, lane width: SDLP). In Fig. 5 (traffic scenarios), defining a metric that could capture human-driving characteristics was more difficult, owing to the complexity that arises due to its dynamic nature. Despite these limitations, as the results show (Figs. 4 and 5), the strength of the cost function (perceived risk) and the risk-threshold driver model lies in the fact that they generate human-like behaviours in different road and traffic conditions, including previously unseen scenarios. Such a generalizable model in which the behaviour emerges from an intrinsically motivated cost does not only provide understanding about human motivations for driving, but also has applications in the design of automated systems. For example, it could be used to make the automated vehicle drive in a human-like manner, which is reported to be preferred by humans[4,63]. Machine learning algorithms could use the 'perceived risk' (cost function) as a feature that could be extracted from demonstrated human-driving trajectories.

Our model has been developed for unassisted driving. However, since its behaviour emerges from the underlying motivations for driver adaptation, we hypothesise that it should be able to capture driver adaptations to various driving support systems. For example, drivers drove faster when their vehicle was equipped with lane-keeping assistance based on HSC than in a car without this assistance[70]. The DRF model should be able to predict this speeding behaviour, since HSC essentially provides a 'channel' on the road through which it guides the driver, reducing the driver's perception-action uncertainty. This would translate to a narrower DRF, which allows a driver to drive faster before exceeding his/her risk threshold. This thought experiment illustrates that a generalizable model in which behaviour emerges from underlying cost functions, not only predicts unassisted driver behaviour but also the effect of automated and assistive technologies (on driver behaviour).

In short, maintaining the 'consequence of the human's perception-actions noise' under a threshold level seems to be the underlying principle for driver's adaptations in speed and lateral position to a wide variety of road and traffic conditions.

## Methods

**Driver model control structure.** This paper focuses on validating the DRF (the dynamic field). However, to generate model predictions on human-driving behaviour, the risk metric calculated using the DRF needs to be connected to a controller that converts the risk metric into control actions. We chose a simple control algorithm over more complex ones for two reasons. First, we wanted to avoid the ambiguity in attributing the driver model's behaviour to the complex algorithm instead of the DRF. Second, we wanted to avoid unnecessary complexity in formalising the optimisation problem. The DRF is an analytically calculable non-linear function (of the driver's states). However, since the environment is represented as a discretized (grid) cost map, the risk metric needs to be calculated numerically. Moreover, we need a controller that maintains the cost above a certain threshold and not one that minimises it. Hence, formulating the optimisation problem with the necessary constraints would itself be a separate study and is beyond the scope of this paper.

The basic control structure (Fig. 7a) includes a driver model that uses the information from the environment and the feedback from the vehicle kinematics to generate control actions ($v_k$: speed, and $\delta_k$: steering angle). The inner workings of the driver model block are shown in Fig. 7b. The DRF is multiplied with the cost map of the driving scene, and summed over all points to provide us with the quantified perceived risk (cost). This cost is then used by the driver model algorithm, which is based on the risk-threshold theory, to generate the control actions.

**Driver model algorithm.** The perceived risk ($C$), in combination with the risk threshold ($C_t$) and desired speed ($V_{des}$), is used to formulate the DRF Model. $V_{des}$ is the speed at which the driver wants to drive on an open straight road, uninhibited.

In accordance with the risk-threshold theory, the model tries to maintain the risk ($C$) below the risk threshold ($C_t$), and hence does not provide a specific trajectory, but rather a range of safe trajectories (satisficing). To avoid the 'bouncing' behaviour exhibited by satisficing controllers (Supplementary Fig. 2), the DRF model is complemented by a heading controller for the steering (Eq. (5)).

$$\delta_{k+1} = \delta_k + k_h(\phi_{road} - \phi_{car}),  \qquad (5)$$

where $\phi_{road}$ and $\phi_{car}$ are the heading of the road and car $t_{lah}$ seconds in the future, respectively. The gain of the heading controller is $k_h$. The predictions about the future position and orientation of the car are made using the 'predicted path' calculations explained earlier in the 'Results' section.

The driver model algorithm (Fig. 7c), at each time step ($k$), compares the risk ($C_k$) to risk threshold ($C_t$), and speed ($v$) to the goal ($V_{des}$). This results in four distinct cases of inequality. We do not consider the equality relations (e.g., if $C = C_t$) because, practically they rarely occur.

(1) If ($C_k < C_t$ and $v_k < V_{des}$): This condition generally occurs when you start the journey. The model speeds up at a rate proportional to ($V_{des} - v_k$). The parameter $k_v$ (specific for each driver) represents how aggressively the model accelerates. The steering is determined by the heading controller ($\delta_{head}$). Hence, $\delta_{k+1} = \delta_{head}$ and $v_{k+1} = v_k + k_v(V_{des} - v_k)$.

(2) Else if ($C_k > C_t$ and $v_k < V_{des}$): In this condition, the incurred risk is more than the threshold ($C_t$), and the goal of desired speed has also not been achieved. In this case, we first check if the steering alone can help the model reduce the risk below the threshold. This check is performed by using the *fmin_bound* function, which finds the steering angle $\delta_{op}$ (within the bounds of $\delta_k - 180°$ to $\delta_k + 180°$) that minimises the risk ($C_k$) assuming a speed of $v_k$. It also calculates the risk ($C_{op}$) at this $\delta_{op}$.

(2a) If the model can find a $\delta_{op}$ such that $C_{op} < C_t$, then we continue to accelerate (to achieve our goal) and steer using $\delta_{opt}$ that reduces $C_k$ to $C_t$ (and not $\delta_{op}$ that reduces $C_k$ to $C_{op}$). This is done so that the model does not 'over correct'. If we were to use $\delta_{op}$ to minimise $C_k$ to $C_{op}$, it would always take the model to the lane centre. Hence the model tries to apply a steering that is just enough to reduce the risk ($C_k$) and get it below the threshold ($C_t$). Hence $\delta_{k+1} = \delta_{opt}$ and $v_{k+1} = v_k + k_v(V_{des} - v_k)$.

(2b) If the model cannot find a $\delta$ such that $C_{op} > C_t$ then the model slows down proportional to $C_{op} - C_k$ (and not $C_{op} - C_t$) since the steering applied $= \delta_{op}$ is expected to reduce $C_k$ to $C_{op}$. This is done so that we do not slow down more than what is required. Hence, $\delta_{k+1} = \delta_{op}$ and $v_{k+1} = v_k + k_{vc}(C_{op} - C_k)$.

(3) Else if ($C_k < C_t$ and $v_k > V_{des}$): In this case the model slows down, while being steered by the heading controller since the risk is lower than the threshold and speed is higher than what is desired. Hence, $\delta_{k+1} = \delta_{head}$ and $v_{k+1} = v_k + k_v(V_{des} - v_k)$.

(4) Else if ($C_k > C_t$ and $v_k > V_{des}$): In this case both the speed and risk are over the desired limits and hence the model slows down while steering with $\delta_{op}$ that minimises $C_k$. Hence $\delta_{k+1} = \delta_{op}$, and $v_{k+1} = v_k + k_{vc}(C_t - C_k) + k_v(V_{des} - v_k)$.

**Parameter estimation.** The parameters of the DRF model were estimated from the experimental data ($n = 1$; 10 trials normal, 10 trials sport driving). The experiment was approved by the Human Research Ethics Committee (HREC)—TU Delft, and a signed informed consent was obtained from the volunteer. The implementation

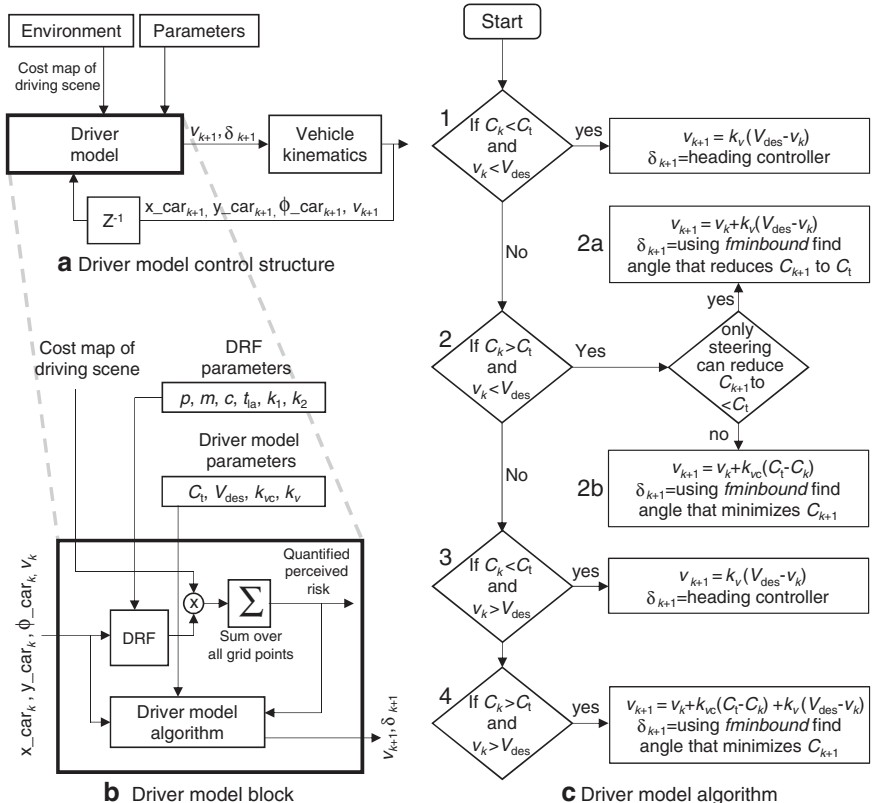

**Fig. 7 Driver model.** A simple driver model that utilises the estimated risk metric to generate control actions is shown. **a** Driver model control structure: The driver model uses the cost map of the driving scene (information about the environment), and the vehicle states (position: $x_{car}$, $y_{car}$; heading: $\phi_{car}$; and speed: ($v$) at $k$th time step to generate the steering angle ($\delta$) and speed ($v$) for $k+1$th time step. **b** The zoomed-in driver model block: The DRF is a dynamic field and changes its shape with vehicle state, which are inputs to the driver model block. The DRF is multiplied with the cost map of the driving scene and summed over all grid points to generate the quantified perceived risk (cost function). The driver model algorithm uses the computed cost function, and the vehicle states to generate the speed ($v$) and steering angle ($\delta$) for next time step. The DRF model algorithm is based on the risk-threshold theory and compares quantified perceived risk ($C$) with risk threshold ($C_t$). The DRF can be individualised based on DRF parameters while the driver model parameters determine how the cost (perceived risk) is converted to control actions (speed and steering). **c** Driver model algorithm: At each time step ($k$), we compare the risk ($C_k$) to risk threshold ($C_t$), and speed ($v_k$) to the goal ($V_{des}$). This results in four distinct cases of inequality.

**Table 1 Driver's Risk Field parameters.**

|  | $p$ | $t_{la}$ | $m$ | $k_1$ | $k_2$ | $c$ |
|---|---|---|---|---|---|---|
| Normal and Sport | 0.0064 | 3.5 | 0.001 | 0 | 1.3823 | 0.5 |

**Table 2 Driver model parameters.**

|  | $C_t$ | $V_{des}$ | $k_{vc}$ | $k_v$ |
|---|---|---|---|---|
| Normal | 3000 | 21.6 | $1.5 \times 10^{-4}$ | 0.14 |
| Sport | 5200 | 26.0 | $1.5 \times 10^{-4}$ | 0.30 |

**Table 3 Driving scene parameters.**

|  | $C_{road}$ | $C_{env}$ | $C_{ovt\ lane}$ | $C_{car}$ |
|---|---|---|---|---|
| Normal and sport | 0 (assumed) | 500 (assumed) | 3.5 | 2500 |

of the track in a fixed base driving simulator is shown in the Supplementary Video 1. Simulations of the DRF model in normal and sport parameter settings are shown in Supplementary Videos 2 and 3.

The parameters can be segregated into three types: first, the DRF parameters that determine the shape of DRF, and are specific to each person. Second, the driver model parameters that connect the risk estimated by the DRF to the control inputs of the vehicle. Third, the environment parameters that describe the consequences of being in a particular state (position, velocity, etc.).

DRF parameters (Table 1): As explained in the 'Results' section, the six parameters ($p$, $t_{la}$, $m$, $c$, $k_1$, $k_2$) define the DRF. Parameter $c$, which represents the initial width of the DRF can be directly calculated from the width of the ego car (2.0 m). The remaining five parameters were estimated using the grid search algorithm.

Driver model parameters (Table 2): The driver model parameters include the speed controller gains ($k_{vc}$, $k_v$), the risk threshold ($C_t$), and the desired speed ($V_{des}$). Parameters $V_{des}$ and $k_v$ can be directly estimated by driving on a long straight section of a wide road, where the driver reaches his/her unbounded desired speed ($V_{des}$) while accelerating (proportional to $k_v$) from a standstill. $k_{vc}$ and $C_t$ were estimated using the grid search algorithm.

Environment parameters (Table 3): The environment parameters define the consequence of being in a particular state (restricted to position, in this study). These parameters are independent of the driver and hence are the same for everyone. Personalised driving behaviour is obtained by changing the parameters

of the DRF and the driver model. In this paper, we assumed the cost (consequence) of being in the 'ego lane' ($C_{road}$) = 0, and outside the lane boundary ($C_{env}$) = 500. The costs of all other objects in the environment were identified relative to $C_{env}$. Different objects have different costs; for example, a car in traffic may be assigned a cost of 4000, and a roadside tree may be assigned a cost of 8000. However, since the focus of this paper is to demonstrate the working of the model, and not identifying the costs of different obstacles, all the obstacles in our simulation were identical: a sedan (1.8-m wide and 5-m long). This 'obstacle car' traversed with different speeds (for overtaking, oncoming and car-following scenarios), or was parked alongside the road (for obstacle avoidance, asymmetric and symmetric road furniture). In all these scenarios, the same cost ($C_{obs}$) was assigned to the car, as identified using the grid search algorithm. The overtaking lane ($C_{ovt\ lane}$) was 'modelled' as rectangular obstacles with a 'very low cost' (identified using grid search), while the oncoming

lane was assumed to be four times as dangerous (four times the cost) as the overtaking lane.

The grid search algorithm tried to minimise $\sum_{i=1}^{3} (y_{i\,\mathrm{model}} - y_{i\,\mathrm{experiment}})^2$, where $i = 1$: steering angle, $i = 2$: speed, $i = 3$: lateral deviation from the lane centre. All the signals were a function of the distance travelled along the lane centre. Tables 1, 2 and 3 report the estimated parameter values for the 'normal' and 'sport' condition. It has to be noted that, to personalise the DRF model to an individual, only seven parameters need to be estimated ($p$, $t_{\mathrm{la}}$, $m$, $c$, $k_1$, $k_2$, $k_{\mathrm{vc}}$ and $C_t$). DRF parameters (Table 1) and the driving scene parameters (Table 3) were estimated only from the 'normal' condition and were used for 'normal' and 'sport' parameter setting of the DRF driver model, since neither the driver nor the driving scene changed. Only the task instruction had changed, due to which (we assume) that the manner in which the driver translates his/her perceived risk into steering and speed-control action changes, which is dictated by the driver model parameters (Table 2).

**Reporting summary**. Further information on research design is available in the Nature Research Reporting Summary linked to this article.

## Data availability
The driving simulator experiment data, the simulation data that support the findings of this study, and the source data for Figs. 3, 4 and 5 are available in the 4TU.Centre for Research Data with the identifier (https://doi.org/10.4121/uuid:8132bccd-e900-4ba0-942e-c3114502bda2).

## Code availability
The DRF Model MATLAB code that supports the findings of this study and a MATLAB GUI that helps explain the DRF are available in the 4TU.Centre for Research Data with the identifiers: DRF model: https://doi.org/10.4121/uuid:ec0f2742-e665-4af9-bf37-8fe1761a8a62 and DRF GUI: https://doi.org/10.4121/uuid:1230ca50-4120-47b2-b6de-35d41c0a4d8a.

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

## Acknowledgements

The Netherlands Organisation for Scientific Research (NWO) funded this project. S.K. and D.A. were supported by the VIDI 14127 project, and J.D.W. was supported by the VIDI 178047 project. We also appreciate the valuable help and feedback provided by S.M. Petermeijer, T. Melman and J.N.P. Giltay.

## Author contributions

S.K. conceived the study; S.K. formulated the model; S.K. piloted the study and collected the data; S.K., J.d.W., D.A. conceived the analysis; S.K. analysed the results and literature; S.K. prepared the figures; S.K., J.d.W., D.A. wrote the paper.

## Competing interests

The authors declare no competing interests.
