## [Peer Review File · Nature Communications]

Reviewers' comments:

Reviewer #1 (Remarks to the Author):

DRF model proposes a very interesting and smart approach to humanize the controlling behavior of autonomous vehicles which may have several safety implications in automated driving. I enjoyed reading this paper. Although the manuscript lacks providing strong supports from experimental data with human subjects, it validates the model with simulated data (+ with 1 human subject) and data available in current literature. The authors also elaborated on the concept of cost function and how it contributes to the human-like driving behavior of the autonomous vehicles, however, their simple control algorithms may miss a large portion of human complex behavior in interaction with different levels of automation and human adaptive behavior when teaming with automation.

Beyond these points, after carefully checking the manuscript regarding the aims and scopes of Nature Communication, this journal is not an appropriate place to publish the manuscript. The author may seek to find other related journals. Please note that re-submitting the improved form of this manuscript still will be out of the aim and scope of this journal.

Thank you

Reviewer #2 (Remarks to the Author):

The study proposed the Driver's Risk Field (DRF) to estimate the driver's perceived risk. A risk-based driver model which is applicable to different scenarios was developed. To verify the effectiveness of the model, the authors compared it with the results of driving behavior studies published in the literature.

Driver's model has been a heavily studied research topic. But the approach of the paper seems interesting and provides novel information. Combining driving behaviors in multiple scenarios is interesting and meaningful. The authors have highlighted the limitations of previous work, and have focused on which are their contributions. The study is interesting and put together well. In order to improve the paper, I have some recommendations:

1. I suggest to consider other factors such as road markings and traffic signs, in addition to roadside furniture and surrounding traffic.
2. The parameter calibration results came from driving simulation experiments, and some of the verification studies also came from driving simulation experiments. So how can you guarantee the validity of the results in real world driving? Or please justify in a more clear and specific way why the authors are sure the results are representative in real scenarios.
3. The limitations of the model, and the future work that can be conducted from the study should be more specific in conclusion section.

Human-like driving behaviour emerges from a risk-based driver model

Response to Referees Letter

Sarvesh Kolekar*, Joost De Winter, David Abbink
Department of Cognitive Robotics
Faculty of Mechanical, Maritime and Materials Engineering (3mE)
Delft University of Technology – The Netherlands

First of all, we would like to thank both the of you, Reviewer1 (**R1**) and Reviewer2 (**R2**), for investing your precious time to read our manuscript and provide valuable feedback. To make this document more legible, we have used the following colour scheme:

BLUE: Comments from you, the reviewers.

BLACK: Authors' response.

GRAY: Text quoted from the manuscript.

We have provided the authors' response (**AR**) immediately following the corresponding comments from you. We hope that this improves the readability and ensures that we address all your comments. We have also uploaded the revised version of the supplementary material, which include:

1. Supplementary information
2. A Matlab GUI to help explore the Driver's Risk Field (DRF) 3D-function
3. The Matlab code for the DRF model
4. The data from the driving simulator experiment and simulations
5. Videos showing the experiment and the simulation

Comments from Reviewer 1

R1₁: DRF model proposes a very interesting and smart approach to humanize the controlling behavior of autonomous vehicles which may have several safety implications in automated driving. I enjoyed reading this paper.

AR: We thank the reviewer for taking the time to read our paper and provide us with valuable feedback.

R1₂: Although the manuscript lacks providing strong supports from experimental data with human subjects, it validates the model with simulated data (+ with 1 human subject) and data available in current literature. The authors also elaborated on the concept of cost function and how it contributes to the human-like driving behavior of the autonomous vehicles, however, their simple control algorithms may miss a large portion of human complex behavior in interaction with different levels of automation and human adaptive behavior when teaming with automation.

AR: We agree. Our driver model was developed and verified for unassisted driving. As such, we did not develop or verify it to describe the interaction of the driver with different levels of automation. However, we do hypothesise that the underlying processes captured by our driver model, would go a long way in describing adaptation to automation (e.g., behavioural adaptation such as speeding when supported by lane-keeping assistance systems). To better clarify this, we have now added the following text to our discussion.

“Our model has been developed for unassisted driving. However, since its behaviour emerges from the underlying motivations for driver adaptation, we hypothesise that it should be able to capture

driver adaptations to various driving support systems. For example, drivers drove faster when their vehicle was equipped with lane-keeping assistance based on HSC, than in a car without this assistance [70]. The DRF model should be able to predict this speeding behaviour, since HSC essentially provides a ‘channel’ on the road through which it guides the driver, reducing the driver’s perception-action uncertainty. This would translate to a narrower DRF, which allows a driver to drive faster before exceeding his/her risk threshold. This thought example illustrates that generalisable models in which behaviour emerges from underlying cost functions can not only model unassisted driver behaviour but also how driver behaviour is affected by automated and assistive technologies.”

Another point highlighted by the Associate Editor related to this comment was a clearer justification for the use of a simple control algorithm. We have added the following text to the Methods section, to more clearly state our motivation to choose a simple control algorithm over a complex one.

“This paper focuses on validating the DRF (the dynamic field). However, to generate model predictions on human driving behaviour, the risk metric calculated using the DRF needs to be connected to a controller that converts the risk metric into control actions. We chose a simple control algorithm over more complex ones for two reasons. First, we wanted to avoid the ambiguity in attributing the driver model’s behaviour to the complex algorithm instead of the DRF. Second, we wanted to avoid unnecessary complexity in formalising the optimisation problem. The DRF is an analytically calculable non-linear function (of the driver’s states). However, since the environment is represented as a discretised (grid) cost map, the risk metric needs to be calculated numerically. Moreover, we need a controller that maintains the cost below a certain threshold and not one that minimises it. Hence, formulating the optimisation problem with the necessary constraints would itself be a separate study and is beyond the scope of this paper.”

R1₃: Beyond these points, after carefully checking the manuscript regarding the aims and scopes of Nature Communication, this journal is not an appropriate place to publish the manuscript. The author may seek to find other related journals. Please note that re-submitting the improved form of this manuscript still will be out of the aim and scope of this journal. Thank you.

AR: We understand the reviewer’s reservations regarding coherence between the aim of our paper and the scope of Nature Communications. However, we think that the quest for uncovering the underlying principles motivating human behaviour is shared by researchers across fields. Our paper also aims to understand human behaviour, be it in the field of driving. Through a cross-disciplinary approach, that uses principles from biology and engineering, we try to explain emergent human behaviour. We ourselves were motivated to investigate ‘*maintaining the consequence of noise below a threshold*’ principle, after coming across the publications of Harris and Wolpert (1998) and Todorov (2004) from the Nature publications group. These publications are from the field of Sensorimotor control and conduct studies with completely different apparatus and scenarios. However, the uniting factor is that the above mentioned papers (among many others) strive to understand the principles governing human behaviour.

Apart from proposing the Driver’s Risk Field, our paper also provides a summary of the important human behaviour adaptations found in the field of driving (e.g., speed in a curve reduces with increasing curvature). This information could be used as test cases for other generalised theories of human behaviour.

Also, keeping in mind the broad spectrum of readership that Nature Communications has, we have ensured that any driving-specific terminology is explained. For example, we include the following text in the manuscript that helps the reader get a feeling for the terminologies used.

Curve-cutting: “Research has shown that drivers exhibit ‘curve-cutting’, that is they do not follow the centreline of the lane but try to increase the effective radius of travel.”

SDLP: “The effect of lane width was examined using the standard deviation of lateral position (SDLP) and speed. SDLP, which represents the swerving behaviour of a car, is reported to increase with lane width, in a simulator study by Godley et al. [37].”

Hence, considering our aim of trying to understand the underlying principle of human behaviour, and the cross-disciplinary approach, we think that our paper is a good fit for Nature Communications.

Comments from Reviewer 2

R2₁: The study proposed the Driver’s Risk Field (DRF) to estimate the driver’s perceived risk. A risk-based driver model which is applicable to different scenarios was developed. To verify the effectiveness of the model, the authors compared it with the results of driving behavior studies published in the literature.

Driver’s model has been a heavily studied research topic. But the approach of the paper seems interesting and provides novel information. Combining driving behaviors in multiple scenarios is interesting and meaningful. The authors have highlighted the limitations of previous work, and have focused on which are their contributions. The study is interesting and put together well. In order to improve the paper, I have some recommendations:

AR: We thank the reviewer for reading our paper, and we see by your summary that you thoroughly understand it. We have used your recommendations and they have helped us considerably to improve our manuscript.

R2₂: I suggest to consider other factors such as road markings and traffic signs, in addition to roadside furniture and surrounding traffic.

AR: We agree that the suggested addition of other factors would be both interesting and useful. However, we chose not to do so in the current manuscript, because human response to traffic rules is not trivial. We do expect that the DRF model can potentially accommodate such an addition. Note that the DRF model responds to a cost map which currently only includes the costs due to physical objects. What we mean is that, an external car is represented by a high cost in the shape of a (rectangle) car. However, the model cannot perceive ‘tactical’ risks which are implied through the context of the scenarios. For example, a traffic light would be modelled, according to the current convention, as a very tall and narrow cylinder adjacent to the road boundary. The ego car would simply go around it. This happens because a traffic light doesn’t directly pose a physical danger due to its presence. It indicates an implied danger that humans consciously perceive and translate into actions. One way to implement this in our model will be to add a roadblock that blocks the entire width of the lane, and its height is dependent on the traffic light colour (say green=0, amber=1000, and red=5000). We have now briefly described this by adding the following text and Fig. 6a to the manuscript. Additionally, we will incorporate your suggestion in our future work, when expanding our DRF model to incorporate tactical costs.

“Implementing a satisficing controller in a potential field has its drawbacks. The model did not return to its lane after overtaking the lead car because it can sense hazard only from physical objects (e.g., cars, road boundary) and cannot perceive the ‘tactical’ risk of being in an oncoming lane, at an intersection, or due to traffic lights (Fig. 6a). However, the structure of the model facilitates the addition of these ‘tactical’ costs to different road elements.”

R2₃: The parameter calibration results came from driving simulation experiments, and some of the verification studies also came from driving simulation experiments. So how can you guarantee the validity of the results in real world driving? Or please justify in a more clear and specific way why the authors are sure the results are representative in real scenarios.

AR: We agree that ideally, more on-road studies would have been used to validate our model (and we are currently in the process of gathering on-road data to do so, unfortunately hampered due to the corona virus). That being said, we have taken great effort to locate on-road studies for all scenario’s and have succeeded in doing so for six out of seven scenario’s (only for roadside furniture we could only find driving simulator studies). Along with that, the driving simulator scenario’s report the same phenomena, although sometimes with different metrics and effect sizes. Based on this, we believe to have provided sufficient evidence that the observed phenomena are representative for real behavioural adaptations.

Therefore, although we cannot guarantee the validity of our results in real-world driving, our model validation provides compelling evidence of the model’s ability to capture well-established, representative driving behaviour phenomena reported both in real-world studies as well as driving simulator studies.

To further answer your questions, and provide readers a better understanding of the validity of the results we have also added Supplementary Tables 1-8 that briefly describe the equipment used by the studies we refer to in the main text for validation in each scenario. Additionally, in the

scenarios where we used a simulator study for validation, we provide an additional on-road study, its experimental details, and the details about the results or conclusions (section no/ Table No/ Figure No.).

An example (Supplementary Table 3) is shown below that provides information regarding the studies used for validating metric 2 (speed) in the Lane width scenario.

Supplementary Table 3: On-road validity for lane width-metric 2 (speed) scenario

	Paper	On-road/ Simulator	Description	Conclusion
2. Lane width Metric 2	The effect of lane width on speed (metric 2) was validated using the driving simulator based study of Liu et al., 2016. The simulator has a motion platform and has been validated for use in research. We also cite the on-road study of Fitzpatrick et al., 2000 who found similar results that validate the DRF model's predictions. However, we used the simulator based study of Liu et al., 2016 since the conditions they used (lane widths) were similar to those simulated for the DRF model.			
	Liu et al., 2016	Simulator	(N=24) The driving simulator of Tongji University - China is a motion-base simulator. A real car is placed in the middle of the experimental cabin as the test vehicle. This simulator has been validated in published literature [1] [2]. 1. Chen, Y.R.; Zheng, S.W. Mechanism analysis of vehicle operating characteristic affected by visual environment of underground road. J. Tongji Univ. (Nat. Sci.) 2013, 41, 1031–1039. 2. Cao, C.; Luo, Y.; Wang, J. Driving simulator validation for research on driving behaviour at entrance of urban underground road. In Proceedings of the International Conference on Transportation Information & Safety, Wuhan, China, 25–28 June 2015; pp. 458–468.	Speed increased with lane width (Fig. 4-2d).
	Fitzpatrick et al., 2000	On-road	(N=100) In this on-road study, the speed data was collected between April 1998 and June 1999 during daylight, off-peak periods, and under dry weather conditions. Speed profiles for approximately 100 free-flowing vehicles were taken at each site (several sites in Texas, USA). Vehicle type was identified by observation. The speed profiles were collected using laser guns positioned on the side of the roadway. Techniques used to hide the technicians from passing motorists include the truck blind and locating behind a tree or bushes were used.	Speed increased with lane width (Fig. 8-6 of Fitzpatrick et al., 2000).

While selecting the studies for validating the DRF model we used the following two criteria which we have mentioned in the text.

“Since no single study fully replicates our scenarios, we chose different studies from literature, to compare with the respective DRF model predictions. Wherever possible we chose a (i) naturalistic driving study, and (ii) in similar conditions as simulated.”

In six out of the seven (four: road scenarios + three: traffic scenarios), we found studies in literature that performed the experiments on-road and in simulators. For the 'roadside furniture' scenario we only found studies performed in simulated environments.

“Note that in this paper we do not compare the ‘microscopic’ trajectories of steering angle, speed, and lateral deviation, but assess the behaviour of the model by comparing trends in certain metrics to those reported in the literature. Six out of the seven scenarios were validated using on-road studies or studies from driving simulators backed by on-road studies (only simulator studies found for roadside furniture: Supplementary Table 1-8).”

Out of the six scenarios, in four of them we found studies that conducted experiments in a real car, and in conditions that were comparable to the simulations of the DRF model. Hence these four scenarios (curve radius, car-following, overtaking, and on-coming traffic) were validated using real-car experiments. In the remaining two scenarios (lane width and on-road obstacles) we chose simulator studies over the on-road tests because the conditions used in these studies were similar to that simulated for the DRF model. Additionally, we also state that the results obtained in the on-road studies are coherent with the selected simulator study.

Lane width: “Lateral position: SDLP, which represents the swerving behaviour of a car, is reported to increase with lane width, in a simulator study by Godley et al. [37]. They examined the SDLPs of participants on three different lane widths (2.5, 3.0, 3.6 m) (Fig. 4-2b). Similar results are reported in other simulator [38][39] and on-road studies [40] which are coherent with the predictions of the DRF model (Fig. 4-2a)”

On-road obstacles: “Several researchers have reported, in on-road studies, that on-street parking induces ‘traffic calming’ by reducing the average speed [45][46][47]. We selected the simulator study of Edquist et al. [48] because they measured the effect of on-street parking on lateral position and speed.”

R2₄The limitations of the model, and the future work that can be conducted from the study should be more specific in conclusion section.

AR: We have added a new figure (Fig. 6) that illustrates the four main limitations of the proposed DRF model and also provide suggestions as to how these limitations can be solved.

“Implementing a satisficing controller in a potential field has its drawbacks. The model did not return to its lane after overtaking the lead car because it can sense hazard only from physical objects (e.g., cars, road boundary) and cannot perceive the ‘tactical’ risk of being in an oncoming lane, at an intersection, or due to traffic lights (Fig. 6a). However, the structure of the model facilitates the easy addition of these ‘tactical’ costs to different road elements. Other limitations include the use of car-kinematic model, using a circular arc for ‘predicted path’ calculations, and the DRF extending only in front of the ego car. In future iterations a car-dynamic model, a spline instead of a circular arc (Fig. 6b), and a DRF that surrounds the vehicle on all four sides (Fig. 6c) can help generate better microscopic trajectories and generate behaviour in more scenarios (e.g., ego car being overtaken).”

Figure 6: **Limitations of DRF model.** (a) Tactical costs: The DRF model can only perceive physical risk from objects such as cars, trees, etc. However, it cannot perceive the risk from potential oncoming traffic which is currently not in its field of view. Hence at an intersection, rather than slowing down, it will speed up since there is larger road-area available, which is contrary to what a human would do. This can be solved by introducing additional ‘tactical costs’ that artificially increase the risk of an intersection (red square). This approach can be extended to other elements such as traffic lights, or zebra crossings. (b) Predicted path: For simplicity, the DRF model currently uses a circular arc for predicting the path (for preview time t_{la} seconds). This circular path arises due to the assumption that the current steering angle (δ) and speed (v) will be held constant over the preview time. However, we can optimise for a vector of steering angles and speed (as is done in a Model Predictive Control). This allows for a flexible DRF and better prediction of microscopic trajectories. (c) Surround DRF: In this paper the DRF only extends in front of the vehicle (top). However, the risk field extends on all four sides. The bottom image is merely a suggestion and the shape has not been investigated. This ‘surround DRF’ will help predict human driving behaviour in additional scenarios such as: being followed by another car, being overtaken, lane change manoeuvres, etc. (d) Uncertainty in dynamic obstacles: The DRF represents the driver’s (self) perception-action uncertainty. However, the motion of dynamic obstacles is less predictable. In this paper, we assumed that the position of the dynamic obstacles was accurately known and hence ignored the uncertainty. But this will have to be accounted for in the future iterations of this model.

Your comments have improved this manuscript even further. Once again, thank you very much for your time and feedback, and we hope you and your families are safe during these unprecedented times.

REVIEWERS' COMMENTS:

Reviewer #2 (Remarks to the Author):

The authors have spent a great efforts in improving the paper. All my comments have been carefully addressed.

Human-like driving behaviour emerges from a risk-based driver model

NCOMMS-20-01946B

(Response to Referees - Letter)

Sarvesh Kolekar*, Joost De Winter, David Abbink
Department of Cognitive Robotics
Faculty of Mechanical, Maritime and Materials Engineering (3mE)
Delft University of Technology
Mekelweg 2, 2628 CD, Delft– The Netherlands

Comments from Reviewer 2

Reviewer 2: “The authors have spent a great efforts in improving the paper. All my comments have been carefully addressed.”

Authors: We are very glad that our revision was in the correct direction and satisfactorily answered your questions. Once again we thank you for reviewing our work and providing us valuable feedback that made our work better.